# Recommendation System for a Delivery Food Application Based on Number of Orders

**Claudia N. Sánchez** [1] , **Julieta Domínguez-Soberanes** [2,*] , **Alejandra Arreola** [3] **and Mario Graff** [4]

1 Facultad de Ingeniería, Universidad Panamericana, Josemaría Escrivá de Balaguer 101, Aguascalientes 20296, Mexico
2 Escuela de Dirección de Negocios Alimentarios, Universidad Panamericana, Josemaría Escrivá de Balaguer 101, Aguascalientes 20296, Mexico
3 Torus Tecnologías SAPI de CV, Venustiano Carranza 104-D, Colonia Centro, Aguascalientes 20000, Mexico
4 CONACYT—INFOTEC Centro de Investigación e Innovación en Tecnologías de la Información y Comunicación, Cto. Tecnopolo Sur 112, Pocitos, Aguascalientes 20326, Mexico
* Correspondence: jdominguez@up.edu.mx

**Abstract:** With the recent growth in food-delivery applications, creating new recommendation systems tailored to this platform is essential. State-of-the-art restaurant recommendation systems are based on users' ratings or reviews, with data that are obtained from questionnaires or online platforms such as TripAdvisor, Zomato, Foursquare, or Yield. However, not all users give ratings or reviews after their purchase. This document proposes a recommendation system whose input is the number of orders stored by a real food-delivery application. These data are always available for all food-delivery applications and are stored all the time. Our proposal is based on the nearest-neighbor technique that calculates the client's preferred restaurants and analyzes other clients with similar buying patterns. In addition, we propose a performance metric that can be used for this specific recommendation system that is based on real restaurant sales. We use a real dataset (available online) to validate our proposal. Based on our experiments, the recommendation system successfully gives only an average of 7.7 options from 187 that are available. We compared our proposal with other state-of-the-art recommendation techniques and obtained a better performance. Our results indicate that it is possible to generate recommendations based on the number of orders, making the use of a restaurant-recommendation system feasible in a real food-delivery application.

**Keywords:** food delivery; recommendation system; nearest neighbors; number of orders

## 1. Introduction

Recently, we have witnessed unprecedented use of technology and its services [1]. The salient feature of the internet is its ubiquity, namely, the network is available at home, at the university, and on small portable devices (phones and watches), sensors, etc. [2]. However, thanks to the growth of internet services, we have a problem of information overload. Recommender systems are algorithms that contribute to the resolution of the problem of information explosion [3]. They have been created to assist people in deciding, from within a large group of options or an overload of information, the one that best suits their personality interests and preferences within a wide range of circumstances [4]. These have been successfully developed in many areas, such as movies [5,6], books [7], education [8], music [9], patient diet [10], and online shopping [11,12], to mention a few.

Therefore, the particular interest of this research is related to food-delivery services since this type of business has increased considerably due to the COVID-19 pandemic [13]. For that reason, it is essential to generate methodologies for restaurant recommendations. The recommendation system helps customers to identify restaurants from an overwhelming group of options by matching customer preferences as much as possible [14]. For restaurants, the recommendation system helps restaurants make free advertising and increase

their turnover [15]. Most food-delivery services are applications designed to be used on mobile devices, where the use of recommendation systems is critical since, with small screens, it is essential that what is shown is genuinely relevant for customers to help them in the decision process [16].

Recommendation systems typically target individuals and are based on individuals' viewing, purchasing, or rating history [17]. Specifically, the documents that propose restaurant recommendation systems are based on ratings or reviews proportioned by customers, using data from own-made apps [15,17,18] or platforms such as TripAdvisor, Zomato, Foursquare, or Yield [14,16,19–23]. However, only some customers gave ratings or reviews after their purchases, making implementing these systems in real food-delivery applications difficult. This document proposes a recommendation system whose input data are real sales transactions, which are always available and stored in all food-delivery applications. Most studies explored restaurant attributes on the overall customer experience (food, service, atmosphere, and value). However, when people buy through applications, they can only enjoy part of the experience in restaurants [24]. The data that all the food-delivery applications have from all customers consist of the history of orders. The number of orders reveals customer preferences. A customer who likes a particular restaurant tends to place several orders from the same place. Most recommendation systems are focused on predicting the rating proportioned by users [21,25,26] and use performance metrics such as accuracy, recall, mean squared error, etc. In this case, we want to predict which restaurants can be preferred by users. For that reason, we propose our own performance metric.

To summarize, this research contributes to presenting a recommendation system for a food-delivery application that uses the number of orders as input. The system uses the nearest clients considering similar buying patterns to make a recommendation. Quite similar to [5,27], our proposal calculates the similarity among customers. To the best of our knowledge, there are no recommendation systems based on the number of sales or orders. In addition to the recommendation system, we propose a methodology for validating this restaurant recommendation. We analyzed the performance of our proposal by using a dataset with real sales from 2019 to 2021, provided by a local food-delivery application (data available online). Then we compared our proposal against matrix factorization [28], a state-of-the-art technique that is widely used in recommendation systems.

The rest of the manuscript is organized as follows. Section 2 presents the literature review. The dataset is described in Section 3. The proposed recommendation system is presented in Section 4. Section 5 presents the experiments and results. The discussion is given in Section 6. Finally, Section 6 concludes this document.

## 2. Literature Review

This section briefly reviews different state-of-the-art recommendation systems tailored to restaurants. Moreover, we present the main techniques that are used in recommendation systems. Finally, we briefly present the k-nearest neighbor methodology, which is the base of our proposal.

### 2.1. Recommendation Systems for Restaurants

This subsection analyzes the documents that are tailored to recommendation systems for restaurants (see a list in Table 1). Some of them use specific users' information. For example, Roy et al. proposed Altered Client-Based Collaborative Filtering (ACCF) [25], a novel collaborative filtering algorithm for grouping recommendations. ACCF employs the Dragonfly Algorithm to deal with the sparsity and neighbor selection. The restaurant-recommendation system was utilized as a testbed to validate ACCF. They used the data of 20 clients from the dataset proposed in [29]. Specifically, they used ratings and users' information, such as if the clients were smokers and their drinking level, activity, and budget. Using users' private data makes their implementation difficult in real life because only some users will proportionate it.

Most of the works are survey-based studies or use data from social networks such as TripAdvisor, Zomato, and Yield, among others. Zhang et al. [14] proposed an approach that combines group correlations and customer preferences, using TripAdvisor data in their experiments. A 5-star rating scale is used on the platform, ranging from 1 ("terrible") to 5 ("excellent"). Chatterjee (2020) [23] uses text-mining techniques to find review sentiment. The research focuses on relationships between quantitative ratings, with a scale like the one above, and information from qualitative textual data taken from TripAdvisor. It used an Artificial Neural Network, Random Forest, and a Support Vector Machine to explain how the reviews help select a hotel and provide better predictive power for the best recommendations. Moreover, Zhang et al. [19] proposed a restaurant decision support model using social information for tourists on TripAdvisor; their experiments used the total ratings, the number of reviews, and the ratings for food, service, and atmosphere. The model introduced fuzzy sets to denote online reviews and used Bonferroni to consider interdependence among criteria. Hartanto and Utama [20] also used fuzzy logic, cosine similarity distance, selection, and optimization to provide a restaurant recommendation system for individual users or groups. Their experiments were based on questionnaires and reviews of 75 restaurants and 8 customers from Zomato. They considered the following features: type of food, location, price, and taste ratings. Asani et al. [21] presented a restaurant-recommendation system based on sentiment analysis. The system gathers users' comments from TripAdvisor and recommends restaurants based on preferences. It uses hierarchical and partitioning clustering to process comments and classify them according to certain similarities. Furthermore, Zhai et al. [22] proposed a methodology to locate the most popular urban restaurants in an area by taking user reviews from restaurants. All the information stated in their research was taken from Dianping, like TripAdvisor. Wang and Yi [15] proposed a restaurant recommendation algorithm for the Chinese app O2O based on the rank-centroid/analytic hierarchy process. They considered food, price, and service factors, using a five-point scale. Some scores were taken directly from the app, but others were extracted from user text evaluations. Worth Eat II [18] is an intelligent application for finding restaurants, with recommendations based on food price, taste rating, and cleanliness rating. All of these factors are also valued on a five-point scale. They used fuzzy logic, Euclidean distance, and hill-climbing to calculate the recommended restaurants. In contrast, Marques et al. proposed BomApetite [16], a mobile system to recommend restaurants providing individual and group recommendations and support for collaborative decision-making. It integrates restaurant information from Zomato, TripAdvisor, Foursquare, Yelp, and Google Places. Their experiments consisted of ten participants, i.e., six males and four females. Withal, Gartrell, et al. proposed SocialDining [17]. This system provides recommendations for small groups of users who want to meet for food or drink at local restaurants. The Social Likelihood Bayesian model is used to compute individual recommendations, and the heuristic group consensus function based on average satisfaction is used to calculate group recommendations. Their data were collected in their app from 31 users over 15 weeks. The disadvantage of those proposals is that only some users have proportionated ratings or reviews after their purchases, making implementing these systems in real applications difficult. In addition, some documents employ the users' perceptions of service [15,18,19,22], which does not apply to food-delivery applications.

To the best of our knowledge, there are no restaurant-recommendation systems based on actual sales or the number of orders. These types of data are essential because they are always registered, and no questionnaires or ratings are needed from customers.

**Table 1.** List of of documents related to restaurants recommendation systems.

| Reference | Data | Data Source | Model | Objective |
|---|---|---|---|---|
| SocialDining proposed by Gartrell et al. [17] | Ratings of users ($n = 31$) over 15 weeks and information of restaurants ($n = 500$) | Own app and Foursquare | Social Likelihood Bayesian | Recommendations for small groups of users who want to meet for food or drink |
| Zhai et al. [22] | Consumer review scores (food, service, and decoration), number of reviews, number of recommendations, evaluation frequency, and geographic-location data for restaurants ($n = 8259$) | Dianping.com | Principal Component Analysis Kernel Density Estimation, Local Moran's I | Locate the most popular urban restaurants |
| BomApetite proposed by Marques et al. [16] | Users' ($n = 10$) votes for restaurants, using a 5-rating scale. Restaurants' data: type of cuisine, cost, distance, opening hours, number of votes, and restaurants' ratings | Zomato, TripAdvisor, Foursquare, Yelp, and Google Places | After voting, five restaurants having the highest overall ratings are recommended | Mobile system to recommend restaurants to a group, based on the preferences of all the group participants |
| Zhang et al. [19] | Overall ratings; number of reviews; and the ratings of food, service, and atmosphere, using a 5-rating scale. Total of 14,562 records related to 451 tourists and 4820 restaurants | TripAdvisor | Fuzzy sets, Bonferroni, Entropy-based similarity measurement | Restaurant decision support model |
| Zhang et al. [14] | Total of 6269 ratings involving 60 restaurants in New York and 1945 customers. The overall rating in a 5-scale was used | TripAdvisor | Probabilistic linguistic term, groups identification, similarity measurement between customer and groups | Restaurant recommendation that combines group correlations and customer preferences |
| Roy et al. [25] | Ratings and users' information of 20 clients. Smokers (binary feature), drinking level, activity, and budget | [29] | Altered Client-Based Collaborative Filtering | Grouping restaurant recommendation |
| Worth eat II proposed by Utama et al. [18] | Food price, taste rating, and cleanliness rating, valued on a five-point scale | Own app | Fuzzy logic, Euclidean distance, and hill-climbing | Application for finding restaurants |
| Wang and Yi [15] | Food, price, and service factors, valued with a five-point scale | Chinese App O2O | Rank-Centroid/Analytic Hierarchy Process | Restaurant recommendation |
| Chatterjee (2020) [23] | Rating scale and text reviews; 40 hotels with 942 observations | TripAdvisor | Artificial Neural Networks, Random Forest, Support Vector Machines | Explain and predict reviews help select a hotel |
| Hartanto and Utama [20] | Questionnaires and reviews of 75 restaurants and 8 customers | Zomato | Fuzzy logic, cosine similarity distance | Restaurant recommendation for individuals or groups |
| Asani et al. [21] | Users' ($n = 100$) text reviews on restaurants | TripAdvisor | Hierarchical and partitioning clustering | Restaurant recommendation system based on sentiment analysis |

### 2.2. Recommendation-System Approaches

Recommendation systems typically target individuals and employ content-based recommendation and collaborative filtering techniques based on the individual's viewing, purchasing, or rating history [17]. Content-Based Filtering (CBF) is a method that uses the similarity between items—in this case, restaurants—to recommend related elements according to the specific users' preferences without considering information from other users (see [5,27,30] for examples). Furthermore, in Collaborative Filtering (CF), the core component is to design a mechanism to predict preferences based on similar groups of users or items [12] (see [20,21,25,31–35] for examples).

Content-Based Filtering (CBF) is commonly used to generate recommendation systems based on their characteristics and users' particular preferences, generating accurate recommendations without considering the ratings and other users' predilections. In other words, this method recommends similar items to those the user has shown a proclivity for [27]. For example, Ali et al. [5] recommended movies to the user, using CBF, by calculating the similarity between film pairs with the Cosine Similarity Measure to narrow down the possible recommendations. Furthermore, Son and Kim [27] used CBF with a multi-attribute network to recommend movies by computing the Dice similarity method to calculate similarities and generate clusters to give the user an accurate recommendation. Song et al. [36] constructed a hybrid recommendation system combined with a content-based recommendation approach, using the Cosine Similarity Measure to calculate similarities between users and food and the Pearson Similarity Method to find similarities among users.

On the other hand, Collaborative Filtering (CF) is often used, and according to Chen [37], it is still in the mainstream of research for this type of system. CF uses similarities between users and items simultaneously to provide recommendations. The two primary areas of CF are neighborhood methods and latent factor models [28]. The first one focuses on computing similarities between items or users and predicts user preferences based on the neighborhood of users or items. Similar to the documents presented in [25,38], our proposal uses neighborhood techniques. In contrast, latent-factor-based methods use known ratings given by users to acquire an approximate model [39]. Those models characterize both items and users by vectors inferred from rating patterns. These models map users and items to a joint latent factor space so that the inner interactions are modeled. Some of the most successful realizations of latent factor models are based on matrix factorization [28]. Koren et al. defined a matrix factorization model as follows. Each item ($i$) is associated with a vector ($q_i \in \mathbb{R}^d$), and each user ($u$) is associated with a vector ($p_u \in \mathbb{R}^d$), where $d$ represents the dimensionality of the latent factor space. The dot product, $\langle \vec{q_i}, \vec{p_u} \rangle$, captures the interaction between the user ($u$) and item ($i$), or, in other words, the user's overall interest in the item's characteristics. This approximates the user's ($u$'s) rating of the item ($i$) denoted by Equation (1). After the recommendation system completes mapping users and items to vectors, it can easily estimate the rating that a user will give to any item by using Equation (1). There are different techniques for calculating the vectors of matrix factorization. For example, some are based on Gradient Descent [39–41], and others on Neural Networks [26,35,42,43].

$$\widehat{r_{iu}} = \langle q_i, p_u \rangle \tag{1}$$

Most recommendation systems, CBF or CF approaches, are tailored to predict the ratings that users give to items. Commonly, these ratings follow a scale, such as a 5-point scale. Their performance metrics can be the ones widely used in regression or classification problems, such as accuracy, mean absolute error, etc. [21,25,26]. In our case, the idea is to predict users' preferred restaurants. This prediction is calculated based on the number of sales. For that reason, we propose our performance metric that can measure how well our recommendation system can identify preferred restaurants.

### 2.3. K-Nearest Neighbors

The k-nearest neighbors (KNNs) [44] algorithm is a learning methodology that can be applied to solve classification, regression, or clustering problems. The main idea is to find the k-nearest neighbors, or samples, to a specific sample. KNN is performed by defining a distance metric. For example, if we define a dimensional space, the metric could be the Euclidean distance. The main advantage of this methodology is that any distance metric can be used; in consequence, KNN can be used for all types of data, for example, images, text, and numeric data. In classification or regression problems, we can predict the value of the new sample by using the labels of the k-nearest neighbors (samples). Equation (2) shows an example of the prediction of a new sample, $y_{new}$, in a regression problem as the mean of the samples' labels, $y_i$, in the neighborhood, $\mathcal{N}$, where k is the number of nearest neighbors. In clustering problems, the groups can be calculated using the neighborhood of samples. KNN performance depends on the value assigned to k. For that reason, it is recommended to test several values.

$$y_{new} = \frac{\sum_{y_i \in \mathcal{N}} y_i}{k} \tag{2}$$

## 3. Description of Data

The data used in this research were proportioned by Torus Technolgías, a local food-delivery company located in Aguascalientes City, Mexico. The data correspond to orders made between January 2019 and June 2021. They contain the number of orders from 187 restaurants and the registers of the 100 top clients with the most orders. This dataset is available online from our GitHub repository: https://github.com/ClaudiaSanchez/DeliveryFoodApplication_RecommendationSystem, accessed on 15 January 2023.

### 3.1. Sales over Time

This kind of data is challenging to manage since it changes as time passes. Figure 1 shows the sales of restaurants from January 2019 to June 2021. We can observe the restaurants with the most and least orders. Restaurant sales can vary over time due to different factors. For example, some new restaurants in the application suddenly became popular, and their sales grew fast. Moreover, external factors affected the number of sales, such as the COVID-19 pandemic or the fact that some restaurants unsubscribed from the application.

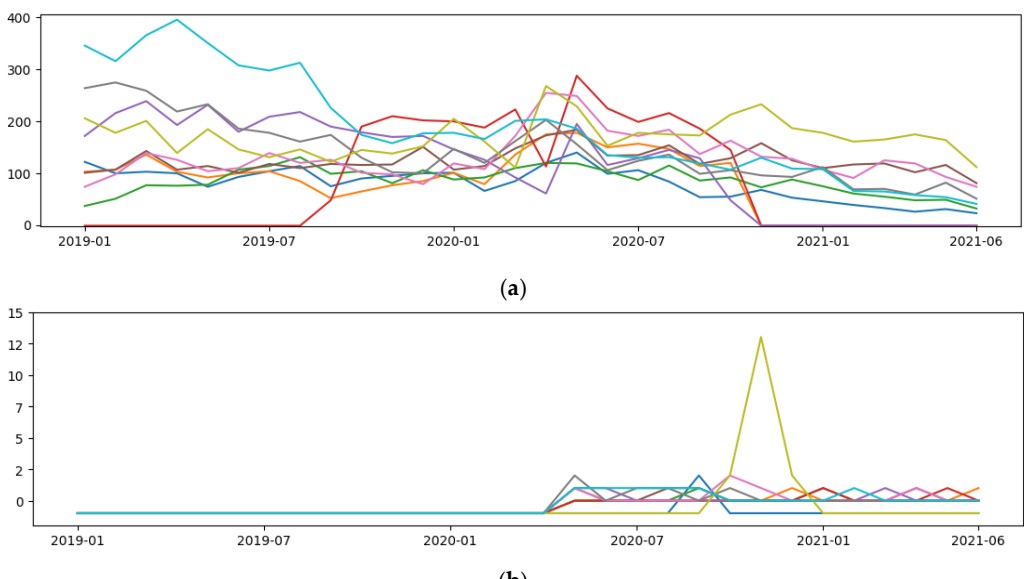

(a)

(b)

**Figure 1.** Sales of restaurants from January 2019 to June 2021. Each line represents one restaurant. The *x*-axis represents months, and the *y*-axis represents the number of orders. (**a**) The top ten restaurants with the greatest number of orders. (**b**) The ten restaurants with the least number of orders.

### 3.2. Preferred Restaurants of the Top Five Clients

Figure 2 shows the number of orders from the five most preferred restaurants from the top five clients. It can be observed that clients buy from different restaurants. For example, Client 102,587 has many orders from Restaurant 27,678. However, other clients in the graph do not have this restaurant as one of their five favorites. Therefore, the number of orders per client varies over time. In the same example, observing Client 102,587 and Restaurant 27,678, we see that orders were more significant in 2019 and 2020 than in 2021. Another example is Client 244,607; she/he placed many orders from Restaurant 138,054 in 2020, but in 2021, she/he placed none. In addition, we have sparse data, meaning that clients order from a few restaurants. In 2019, customers ordered from 15.1 restaurants, on average, with a standard deviation of 6.57. In 2020 and 2021, the average was 16.75 and 7.53, respectively, with a standard deviation of 7.53 and 5.12. This indicates that clients know only some restaurants, and they can explore more options with an appropriate recommendation system.

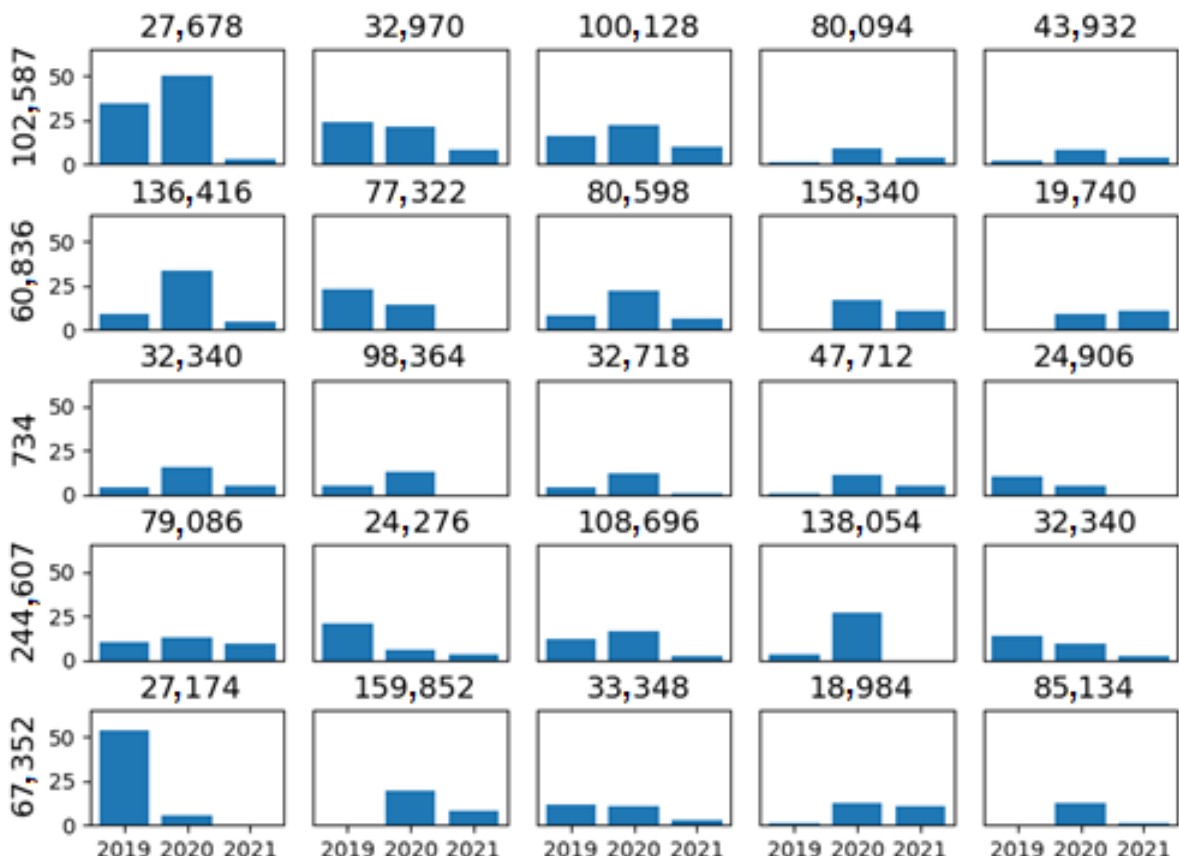

**Figure 2.** The number of orders (*y*-axis) to the five preferred restaurants of the top five clients. Each row represents a client; the ID appears on the left. At the top of each graph is the restaurant ID. The three bars represent the number of orders for 2019, 2020, and 2021.

### 3.3. Groups of Clients and Restaurants

To visually identify groups of clients and restaurants, we use hierarchical clustering based on Euclidean distance, with Ward's minimum variance criterion. Figures 3–5 show the dendrogram representing groups of clients and restaurants in 2019, 2020, and 2021, respectively. When analyzing Client 743 in 2019, he/she appears to be in the last red group, and his/her nearest client is 243,419. In 2020, he/she is in the last red group, and his/her closest client is 463. Finally, in 2021, he/she appears in the seventh green group, whose nearest client is 16,655. Here we confirmed that groups of clients and neighborhoods

change over time. In addition, a recommendation system is possible by using this kind of data because groups can be clearly identified.

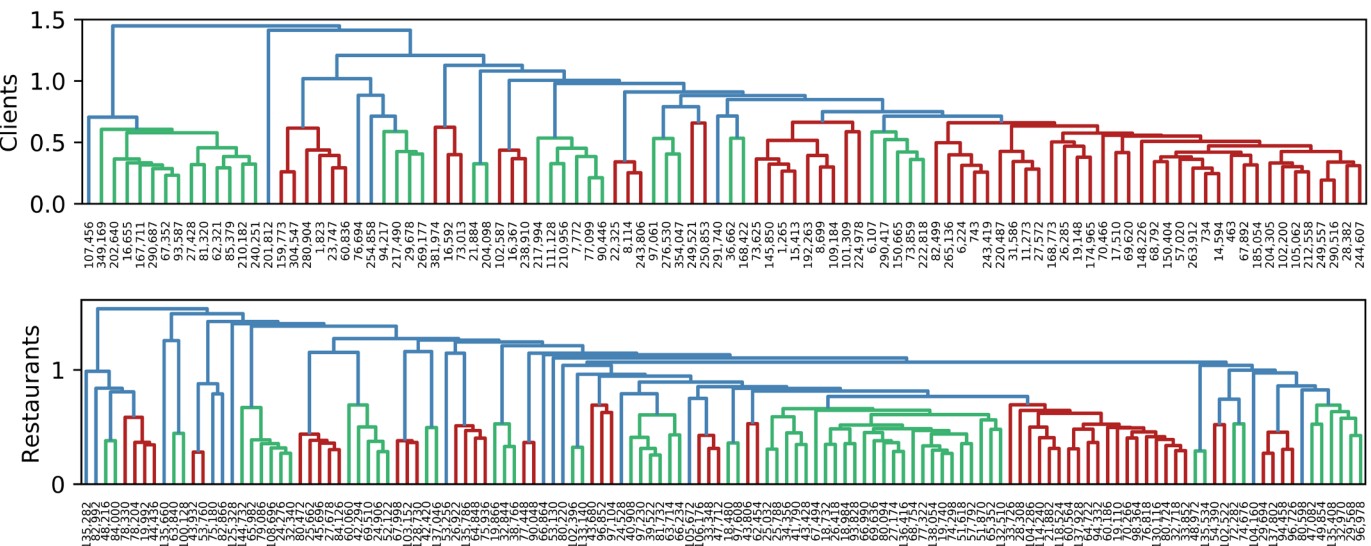

**Figure 3.** Similarity among clients and restaurants in 2019. The length of vertical lines represents the distance between items. The shorter the line, the closer the items. Different colors represent possible groups of items.

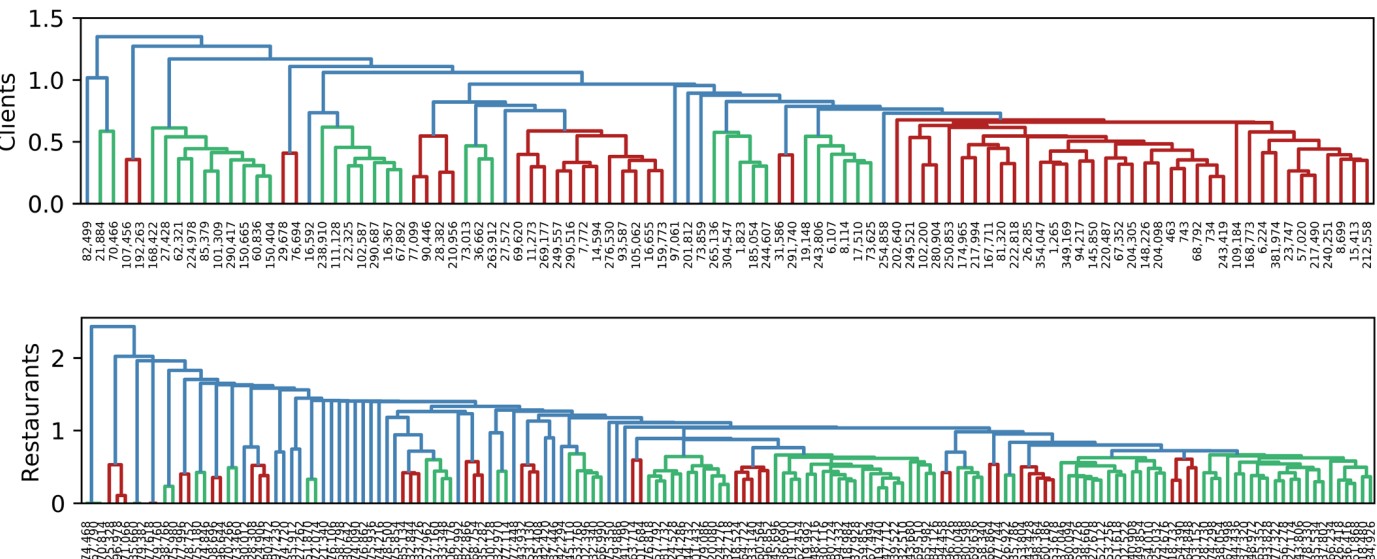

**Figure 4.** Similarity among clients and restaurants in 2020. The length of vertical lines represents the distance between items. The shorter the line, the closer the items. Different colors represent possible groups of items.

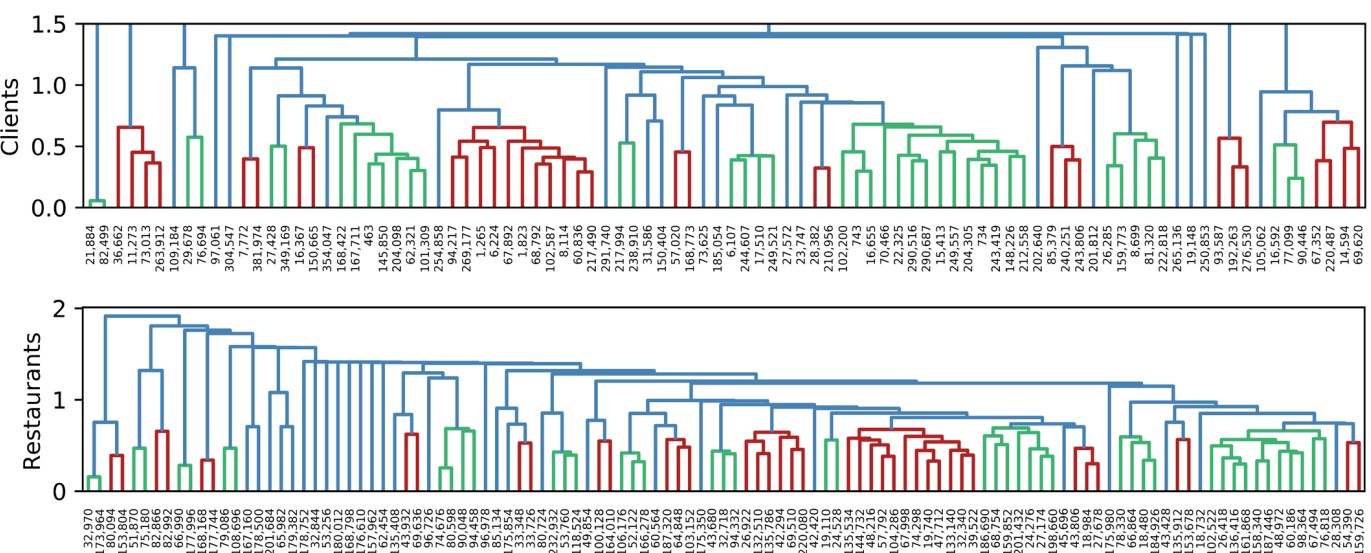

**Figure 5.** Similarity among clients and restaurants in 2021. The length of vertical lines represents the distance between items. The shorter the line, the closer the items. Different colors represent possible groups of items.

## 4. Proposed Recommendation System

This research aims to generate a recommendation system for a food-delivery application based on the number of orders. In the previous section, we described the patterns of the data. We noticed that we need to handle issues to generate a custom recommendation system. First, we have a sparse dataset; clients tend to buy from a few restaurants, and most of the values in the client vectors are zero. Second, the data vary over time; for example, the preferred restaurants of a client could be different in the next year, or the restaurants get in and get out of the application.

We propose a recommendation system based on the k-nearest neighbor technique. For calculating the recommendation of a client, we use the k-nearest clients; in other words, those clients with the nearest consuming pattern. For making the recommendation based on the last number of orders, we recommend using the data of the last semester or year. In our experiments, the data are divided according to year.

### 4.1. Clients and Restaurants Vectors

For analyzing patterns and calculating distances among clients or restaurants, we represent them as vectors containing their number of orders. The vector $c^i$ represents the $i$-th client. Its entries correspond to the number of orders to every restaurant. On the other hand, the vector $b^j$ represents the $j$-th restaurant, whose values correspond to the number of orders from each client. It is essential to note that some clients could have the same preferences for restaurants, but their number of orders differs. Imagine the hypothetical case of having selected only eight restaurants, and the number of orders of Clients 1 and 2 are represented by the vectors $c^1 = \{12, 0, 0, 0, 0, 20, 0, 4\}$ and $c^2 = \{6, 0, 0, 0, 0, 10, 0, 2\}$, respectively. Those clients have the same preferences, meaning that their preferred restaurant is the sixth, followed by the first. The preference is the same, but the number of orders is different. Considering those cases, and for identifying clients' similarities, we use normalized vectors (see Equation (3)). In the example, $\widehat{c^1} = \{0.333, 0, 0, 0, 0, 0.555, 0, 0.111\}$ and $\widehat{c^2} = \{0.333, 0, 0, 0, 0, 0.555, 0, 0.111\}$.

$$\widehat{c^i} = \frac{c^i}{\sum_m c_m^i} \tag{3}$$

### 4.2. Clients' Preferred Restaurants

The recommendation process is explained with a hypothetical example (see Table 2) of six clients ($x$, $o$, $p$, $q$, $r$, and $s$) and four restaurants ($a$, $b$, $c$, and $d$). First, we calculate all the clients' normalized vectors (see Equation (3)). Entries of normalized vectors correspond to the percentage of orders. All values in those vectors altogether must sum to 1.0 (see the rows in Table 2). The minimum percentage of orders, *pmin*, was used to determine the preferred restaurants for each client. Suppose the percentage of orders of client $x$ to a specific restaurant, $a$, is more than *pmin*. In that case, we consider that restaurant $a$ is inside the preferences of client $x$. We recommend using *pmin* $= 0.1$, meaning that restaurants with more than 10% of the orders are preferred. In the next section, we experiment with different values of *pmin*. In our example (see Table 2), using *pmin* $= 0.1$, the preferred restaurants to client $x$ are $a$ and $b$ since the percentage of orders are 0.19 and 0.80, respectively. A real example is shown in Appendix A.

**Table 2.** Example of the proposed recommendation system. Rows represent the clients' normalized vectors ($x$, $o$, $p$, $q$, $r$, and $s$). Columns represent restaurants ($a$, $b$, $c$, and $d$). Blue columns represent the preferred restaurants of client $x$. Green cells represent the recommended restaurant based on our recommendation system.

| | $a$ | $b$ | $C$ | $d$ | Difference with $x$ |
|---|---|---|---|---|---|
| $x$ | 0.19 | 0.01 | 0.80 | 0.00 | |
| $o$ | 0.03 | 0.03 | 0.91 | 0.03 | $(\lvert 0.19 - 0.03 \rvert + \lvert 0.80 - 0.91 \rvert)/2 = 0.135$ |
| $p$ | 0.06 | 0.09 | 0.50 | 0.35 | $(\lvert 0.19 - 0.06 \rvert + \lvert 0.80 - 0.50 \rvert)/2 = 0.215$ |
| $q$ | 0.29 | 0.05 | 0.21 | 0.45 | $(\lvert 0.19 - 0.29 \rvert + \lvert 0.80 - 0.21 \rvert)/2 = 0.345$ |
| $r$ | 0.30 | 0.15 | 0.15 | 0.40 | $(\lvert 0.19 - 0.30 \rvert + \lvert 0.80 - 0.15 \rvert)/2 = 0.380$ |
| $s$ | 0.00 | 0.92 | 0.03 | 0.05 | $(\lvert 0.19 - 0.00 \rvert + \lvert 0.80 - 0.03 \rvert)/2 = 0.480$ |

### 4.3. Calculation of Nearest Clients and Recommendation

Once we calculate the preferred restaurants of client $x$, we can calculate his/her nearest neighbors. We define the distance between client $x$ and the remaining as the average absolute difference between the normalized vectors, but using only the entries corresponding to the preferred restaurants of client $x$. Equation (4) shows the metric used as distance; $\widehat{c^x}$ and $\widehat{c^j}$ represent the normalized vector of clients $x$ and $j$, respectively, and $P_x$ is the set of entries corresponding to the preferred restaurants of client $x$. In our example, the distance between client $x$ and client $o$ is $(\lvert 0.19 - 0.03 \rvert + \lvert 0.80 - 0.91 \rvert)/2$. Using $k = 3$, the k-nearest clients of $x$ are $o$, $p$, and $q$. Finally, the recommended restaurants for client $x$ correspond to the preferred of the k-nearest clients. These correspond to the ones with a value greater than *pmin* in some of the normalized vectors from the k-nearest clients. To simplify and reduce the number of recommendations, we recommend only the restaurants that were not previously some of the preferred ones for client $x$. In our example, we recommend restaurant $d$ since it has values greater than 0.1 in the normalized vectors of the nearest clients:

$$d\left(\widehat{c^x}, \widehat{c^j}\right) = \frac{\sum_{m \in P_x} \left| \widehat{c_m^x} - \widehat{c_m^j} \right|}{\sum_{m \in P_x} 1} \tag{4}$$

### 4.4. Proposal's Algorithm

Algorithm 1 shows the pseudocode of our proposed recommendation system based on KNN. Our method has two parameters: the number of neighbors, $k$; and the minimum percentage of orders to consider a restaurant among the preferred ones, *pmin*. We recommend using $k = 5$ and *pmin* $= 0.1$. In the next section, we experiment with different values of these parameters.

**Algorithm 1.** Pseudocode of the proposed recommendation system.

**Input:** Historic number of orders from clients to restaurants, represented as clients' vectors, $c^j$, and the vector of the client who wants a recommendation, $c^x$.
**Output**: List of recommended restaurants.
**Hyperparameters**: $k$ and *pmin*.

1.  Normalize clients' vectors:

$$\widehat{c^i} = \frac{c^j}{\sum_m c_m^j}$$

2.  Calculate the set of preferred restaurants of client $x$, $P_x$ which correspond to the entries of $\widehat{c^x}$ that are greater than *pmin*.
3.  Calculate the distance between client $x$ and the remaining:

$$d\left(\widehat{c^x}, \widehat{c^j}\right) = \frac{\sum_{m \in P_x}\left|\widehat{c_m^x} - \widehat{c_m^j}\right|}{\sum_{m \in P_x} 1}$$

4.  Select the k-nearest clients.
5.  Find the entries (restaurants) in the normalized vectors of the k-nearest clients whose values are greater than *pmin*.
6.  Recommend the restaurants found in Step 4 that are not in $P_x$

## 4.5. Metric for Calculating the Recommendations' Performance

Most documents calculate the performance of recommendation systems based on metrics that are used for classification or regression models, such as accuracy, recall, mean absolute error, root mean square error, etc.; examples include [21,25,26]. In those cases, the aim is to measure how well the prediction of ratings is performed. However, in our case, our goal is to predict new preferred restaurants. For that reason, we propose the following methodology for calculating the recommendation performance (see Algorithm 2). First, we remove one of the preferred restaurants of a specific client (Algorithm 2, Steps 3, 4, and 5). Then we normalize the client's vector (Algorithm 2, Step 6) and calculate the k-nearest clients (Algorithm 2, Steps 7 and 8) and the recommended restaurants (Algorithm 2, Steps 9 and 10). If the removed restaurant is one of the recommended ones, we consider it to be a correct recommendation; otherwise, it is an incorrect one (Algorithm 2, Step 11). Finally, we repeat this process for all clients in the dataset and calculate the percentage of clients with correct restaurant recommendations.

**Algorithm 2.** Pseudocode of the performance metric.

**Input:** Historic number of orders from clients to restaurants, represented as clients' vectors, $c^j$, and the vector of the client who wants a recommendation, $c^x$.
**Output**: 1 if the recommendation was successful, or 0 otherwise.
**Hyperparameters**: $k$ and *pmin*.

1.  Normalize clients' vectors:

$$\widehat{c^i} = \frac{c^j}{\sum_m c_m^j}$$

2.  Calculate the set of preferred restaurants of client $x$, $P_x$ which correspond to the entries of $\widehat{c^x}$ that are greater than *pmin*.
3.  Randomly choose a restaurant, $r$, of $P_x$.
4.  Remove $r$ from $P_x$.
5.  Set the entry of $c^x$ corresponding to r as 0:

$$c^x = 0$$

6.  Recalculate the normalized vector of x:

$$\widehat{c^x} = \frac{c^x}{\sum_m c_m^x}$$

| **Algorithm 2.** *Cont.* |
| --- |

7.   Calculate the distance between client $x$ and the remaining:

$$d\left(\widehat{c^x}, \widehat{c^j}\right) = \frac{\sum_{m \in P_x} \left|\widehat{c_m^x} - \widehat{c_m^j}\right|}{\sum_{m \in P_x} 1}$$

8.   Select the k-nearest clients.
9.   Find the entries (restaurants) in the normalized vectors of the k-nearest clients whose values are greater than *pmin*.
10.  Calculate the set of recommended restaurants, $P_r$, found in Step 9 that are not in $P_x$.
11.  Return 1 if $r$ is in $P_r$, and 0 otherwise.

## 5. Experiments and Results

In this section, we analyze the performance of our proposed recommendation system by using different values for *pmin* and $k$ parameters (see Section 4). We compare our proposal against the state-of-the-art collaborative filtering technique based on matrix factorization (see Section 2.2). Finally, we present an analysis of the real application of our proposal.

### 5.1. Number of Preferred Restaurants According to the Value of Parameter Pmin

Since our metric for calculating the recommendations' performance removes one of the preferred restaurants for each client, it is important to know the number of preferred restaurants is modified when varying the parameter of *pmin* (see Figure 6 and Table 3). It can be observed that by using 5% as the minimum percentage for considering a restaurant as one of the favorites, we have around five restaurants among the client's preferences. As expected, if the value of *pmin* increases, the number of preferred restaurants decreases. If we consider a minimum percentage of 10% (as recommended), we have around 2.6 preferred restaurants. Based on the dataset used for this research, we do not recommend a value of *pmin* greater than 0.1 because the number of preferred restaurants is very small ($\leq 2$). In addition, values smaller than 0.1 indicate that we need less than 10% of our orders to consider a restaurant to be one of our favorites.

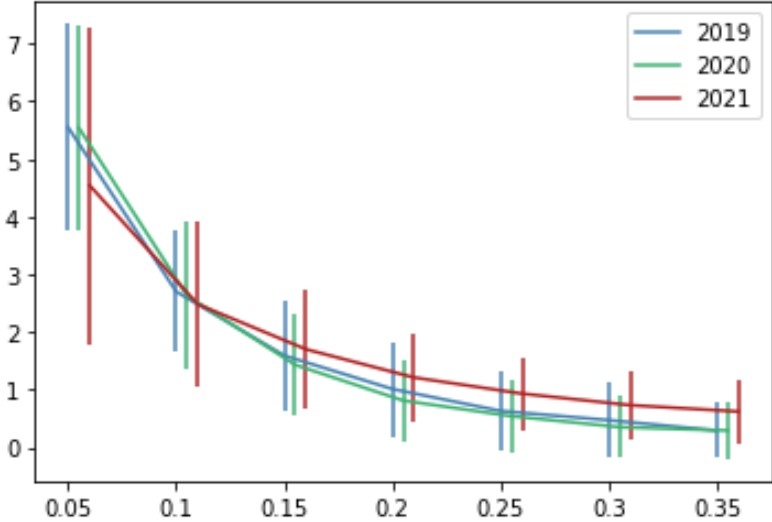

**Figure 6.** The average number of preferred restaurants (*y*-axis) according to the value of the parameter *pmin* (*x*-axis). Vertical lines represent the standard deviation.

**Table 3.** Average number of preferred restaurants according to the value of the parameter *pmin*.

| Year | *pmin* | | | | | |
|---|---|---|---|---|---|---|
| | 0.05 | 0.10 | 0.15 | 0.20 | 0.25 | 0.35 |
| 2019 | 5.55 | 2.70 | 1.59 | 1.01 | 0.63 | 0.29 |
| 2020 | 5.54 | 2.63 | 1.43 | 0.81 | 0.54 | 0.29 |
| 2021 | 4.54 | 2.48 | 1.70 | 1.21 | 0.93 | 0.62 |

*5.2. Recommendations' Performance of Our Proposed Recommendation System*

We tested different values of our recommendation system's parameters: *pmin* and *k*. Figure 7 shows the percentage of correct recommendations using different values for the parameters *pmin* and *k*. To clarify those results, Tables 4 and 5 show the average values of the correct recommendation percentage. On the other hand, Figure 8 and Tables 4 and 5 show the number of recommended restaurants based on the values of both parameters.

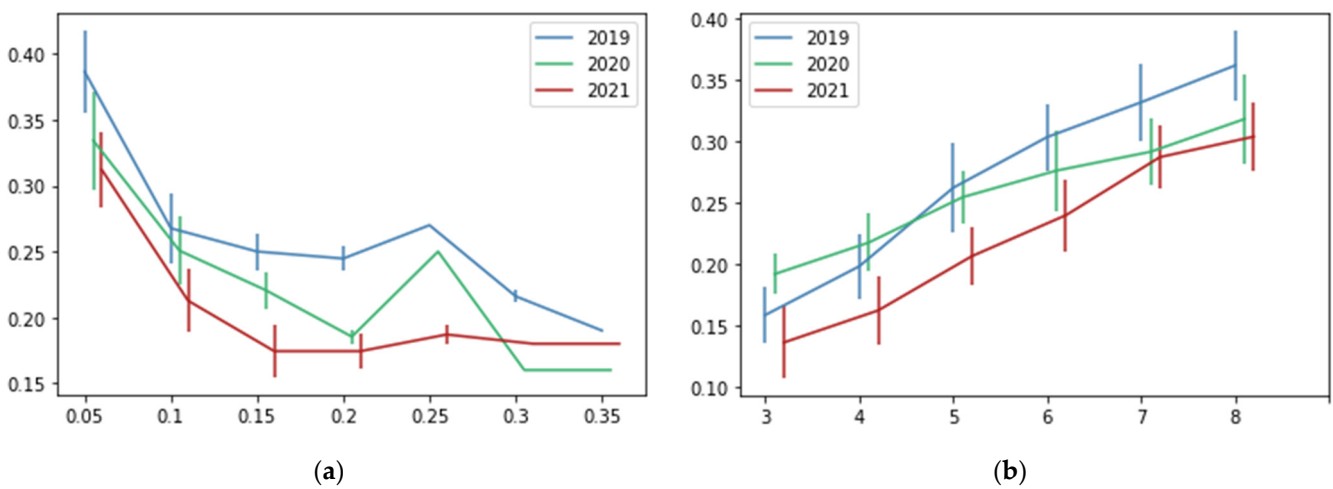

(**a**)  (**b**)

**Figure 7.** Percentage of correct recommendations (*y*-axis) in a scale from 0 to 1, according to the values of the parameters (*x*-axis). Vertical lines represent the standard deviation. (**a**) Testing different values for *pmin*, *k* = 5. (**b**) Testing different values for *k*, *pmin* = 0.1.

**Table 4.** Results of our proposed recommendation system according to the value of the parameter *pmin*.

| Year | *pmin* | | | | | | |
|---|---|---|---|---|---|---|---|
| | 0.05 | 0.10 | 0.15 | 0.20 | 0.25 | 0.30 | 0.35 |
| | Average of the percentage of correct recommendations | | | | | | |
| 2019 | 0.39 | 0.27 | 0.25 | 0.24 | 0.27 | 0.22 | 0.19 |
| 2020 | 0.33 | 0.25 | 0.22 | 0.19 | 0.25 | 0.16 | 0.16 |
| 2021 | 0.31 | 0.21 | 0.17 | 0.17 | 0.19 | 0.18 | 0.18 |
| | Average of the number of recommended restaurants | | | | | | |
| 2019 | 16.31 | 7.58 | 3.77 | 1.79 | 0.85 | 0.57 | 0.39 |
| 2020 | 16.83 | 7.74 | 3.58 | 1.31 | 0.74 | 0.45 | 0.28 |
| 2021 | 15.57 | 7.78 | 4.89 | 2.69 | 1.77 | 1.16 | 0.90 |

**Table 5.** Results of our proposed recommendation system according to the value of the parameter *k*.

| Year | K | | | | | |
|------|------|------|------|------|------|------|
| | **3** | **4** | **5** | **6** | **7** | **8** |
| | Average of the percentage of correct recommendations | | | | | |
| 2019 | 0.16 | 0.20 | 0.26 | 0.30 | 0.33 | 0.36 |
| 2020 | 0.19 | 0.22 | 0.25 | 0.28 | 0.29 | 0.32 |
| 2021 | 0.14 | 0.16 | 0.21 | 0.24 | 0.29 | 0.30 |
| | Average of the number of recommended restaurants | | | | | |
| 2019 | 4.55 | 6.07 | 7.60 | 8.98 | 10.36 | 11.89 |
| 2020 | 4.72 | 6.09 | 7.71 | 9.25 | 10.55 | 11.97 |
| 2021 | 4.66 | 6.23 | 7.74 | 9.34 | 10.89 | 12.34 |

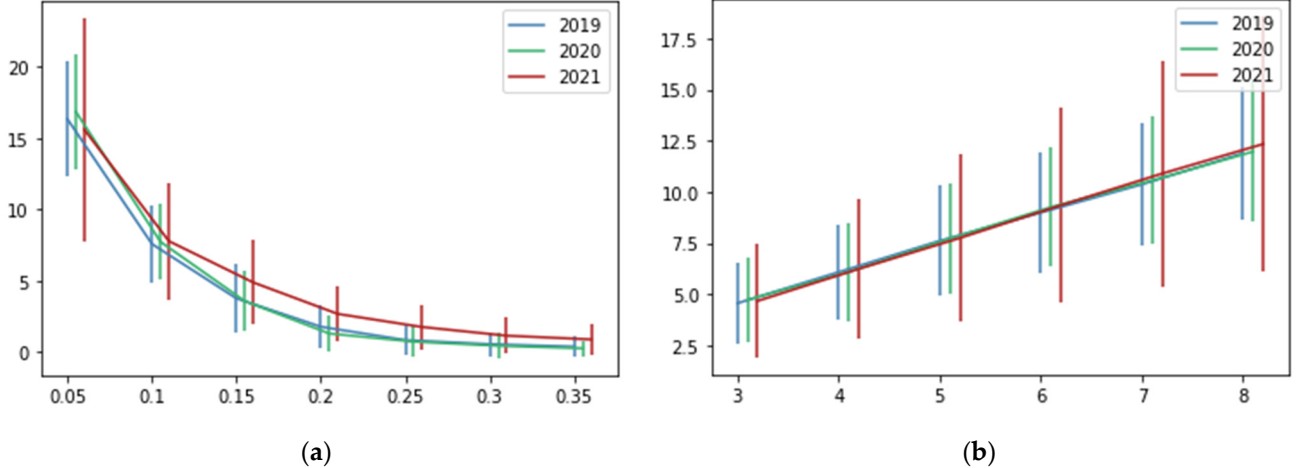

(**a**)                       (**b**)

**Figure 8.** The average number of recommended restaurants (*y*-axis) according to the values of the parameters (*x*-axis). Vertical lines represent the standard deviation. (**a**) Testing different values for *pmin*, *k* = 5. (**b**) Testing different values for *k*, *pmin* = 0.1.

Figure 7 and Table 4 show that when the value of *pmin* increases, the performance of the recommendation system decreases. Moreover, Figure 8 shows that when the value of *pmin* increases, the number of recommended restaurants decreases. Those patterns were expected since the bigger the value of *pmin*, the greater the percentage of orders needed to consider a restaurant a preferred one. An opposite behavior can be observed in the *k* parameter (see Figures 7 and 8 and Table 5). In this case, the bigger the value of *k*, the greater the number of recommendations. In consequence, the better the recommendation system's performance. This is logical because the more clients are considered neighbors, the more chances to obtain restaurants with values bigger than *pmin* for recommendations. We need to find a balance of *k* because, if we consider a big value, the number of recommended restaurants increases. In a recommendation system, the main idea is to obtain only a few recommendations from a wide range of options. In this case, we set *k* = 5 and *pmin* = 0.1, with an average of 7.7 recommended restaurants and a percentage of correct recommendations of around 0.24.

The performance analysis of our recommendation system with the recommended values of parameters (*pmin* = 0.1 and *k* = 5) is presented as follows. The percentage of correct predictions is around 24%. This number may seem small, and the performance of our recommendation system may seem poor. However, Figure 8 and Table 4 show that the average of recommended restaurants is around 7.7, which means that our recommendation system gives the client an average of 7.7 options from 187 available. Moreover, based on our methodology for validation, users have 2.6 preferred restaurants on average. We removed one of them, and 24% of the time our recommendation system can recover it from

187 options, using only around 7.7 recommendations. Based on the number of restaurants and recommendations, we consider this a satisfactory performance.

### 5.3. Comparison of Our Proposed Recommendation System and a Collaborative Filtering Technique Based on Matrix Factorization

As we mentioned in Section 2, matrix factorization is a state-of-the-art technique used for recommendation systems. It creates an embedding space representing users and items to calculate user preferences. The main idea is to combine the data of all users and items to make recommendations. We compared this technique based on matrix factorization against our proposal based on k-nearest neighbors.

Matrix factorization was implemented for calculating a 20-dimensional embedding space. Gradient Descent with a learning rate of 1.0 was used as the optimization algorithm, and only the non-zero inputs in the data were used for training the models. We use the percentage of orders (but only the non-zero inputs) to train the model and measure its performance. Based on the real and the predicted values, we obtained an average mean square error equal to 0.03 with a standard deviation of 0.0012. Knowing that the values are percentages between 0 and 1, 0.03 could be considered a minor error.

Figure 9 and Table 6 show the results of the matrix factorization technique based on the percentage of correct recommendations and the number of recommended restaurants. It can be observed that matrix factorization obtains a bigger percentage of correct recommendations than our proposal. However, the number of recommended restaurants with matrix factorization is significantly bigger than our proposal. When setting the *pmin* to 0.1, the matrix factorization recommends around 66.5 restaurants from 187 options, in contrast to our proposal, which recommends around 7.7 restaurants. However, the main objective of a recommendation system is to filter the data to recommend only a few items to users.

**Table 6.** Results of matrix factorization according to the value of the parameter *pmin*.

| Year | *pmin* | | | | | | |
|---|---|---|---|---|---|---|---|
| | **0.05** | **0.10** | **0.15** | **0.20** | **0.25** | **0.30** | **0.35** |
| | Average of the percentage of correct recommendations | | | | | | |
| 2019 | 0.45 | 0.37 | 0.19 | 0.09 | 0.03 | 0.02 | 0.00 |
| 2020 | 0.44 | 0.32 | 0.17 | 0.05 | 0.02 | 0.01 | 0.01 |
| 2021 | 0.39 | 0.32 | 0.24 | 0.14 | 0.06 | 0.03 | 0.01 |
| | Average of the number of recommended restaurants | | | | | | |
| 2019 | 86.68 | 73.42 | 38.20 | 17.93 | 5.51 | 2.43 | 0.00 |
| 2020 | 85.00 | 62.04 | 32.03 | 10.06 | 4.02 | 1.05 | 0.30 |
| 2021 | 74.38 | 64.27 | 45.83 | 24.85 | 11.75 | 4.87 | 1.71 |

Considering the methodologies for recommendation systems, we used a classical nearest-neighbor technique based on the percentage of orders. Koren et al. affirmed that latent models deliver accuracy superior to classical nearest-neighbor methods [28]. However, for this specific case, the comparison of our proposal based on nearest neighbor performs better than the latent model based on matrix factorization. Tables 4 and 6 show that the average percentage of correct recommendations (using *pmin* = 0.1) with our proposal is 24% and 32% with matrix factorization. We can assume that matrix factorization gives better results. However, in the same tables, we can see that the average number of recommended restaurants (using *pmin* = 0.1) is 7.7 in our proposal and 66.6 in matrix factorization. Considering that the number of restaurants in our experiment is 187, the recommendation of 66.6 restaurants is useless. In contrast, the recommendation of 7.7 restaurants sounds more reasonable. Most of the current recommendation systems that use nearest-neighbor techniques use cosine similarity distance [5,20,21], like our proposal of using normalized vectors. In both cases, the main idea is to use only the direction of vectors and ignore the

magnitude. It allows us to analyze consumption patterns without caring about the number of orders.

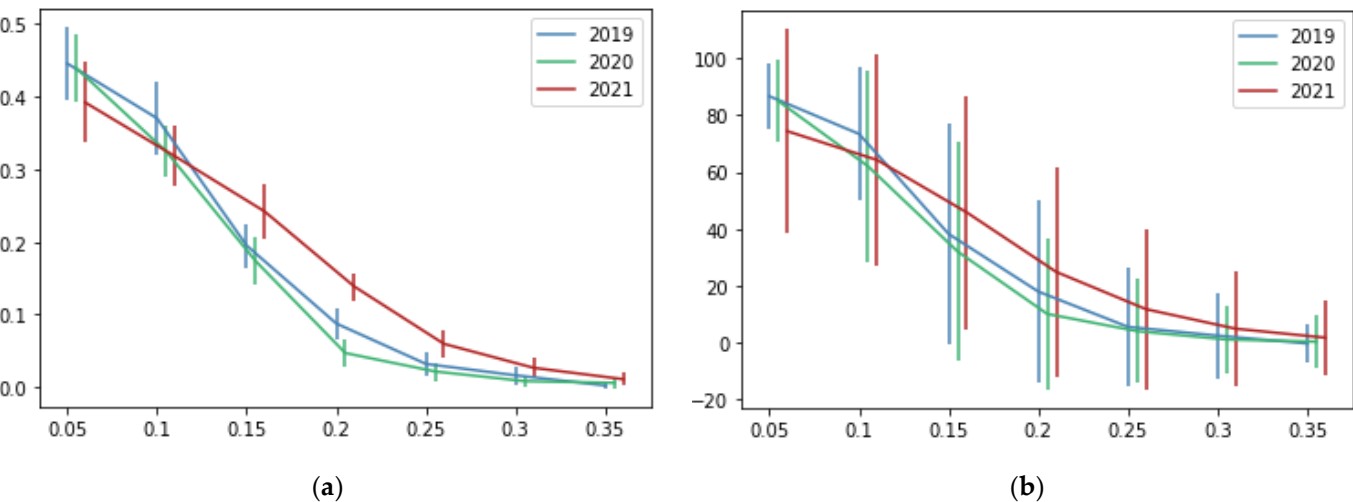

(**a**)  (**b**)

**Figure 9.** Results of matrix factorization according to the values of the parameters *pmin* (*x*-axis). Vertical lines represent the standard deviation. (**a**) Percentage of correct recommendations (*y*-axis) on a scale from 0 to 1. (**b**) The average number of recommended restaurants (*y*-axis).

### 5.4. Analysis of Real Application

The main advantage of this proposal is that it could be applied in a real food-delivery application. Unlike other restaurant-recommendation systems, whose input data are ratings or reviews obtained from surveys [15,18,25,29] or platforms such as TripAdvisor, Zomato, and Yield [14,16,19–22], our proposal directly uses the number of orders that is stored in the application daily. This allows us to have the training data available all the time, and no questionnaires or ratings are needed from customers. To the best of our knowledge, this is the first time a recommendation system for a delivery food application is based on the number of orders.

The model's training could be executed periodically, e.g., weekly. It consists of calculating customers' preferred restaurants and the nearest clients. This is commonly executed in a server and does not affect the application's functioning. Once we have the model, the restaurant recommendation is calculated almost instantly. In this sense, it is cost-efficient because the profits of successful restaurant recommendations can absorb the implementation cost.

### 6. Conclusions

This document proposed a recommendation system for a delivery food application based on orders. Our system uses the sales transactions automatically stored by the application, and no questionnaires or ratings are needed from customers. The methodology used for the recommendation is a nearest-neighbor technique that is based on the percentage of orders. The recommendations are based on the preferred restaurants. We defined a preferred restaurant as one with at least 10% of the clients' orders. In our experiments, we used actual data from a local delivery food application with 187 restaurants and 100 customers in Aguascalientes, Mexico. Clients have, on average, only 2.6 preferred restaurants, and 24% of the time, our system can identify one of those preferred restaurants from a list of 7.7 recommended ones (from 187 available). These numbers confirmed to us that our system has a satisfactory performance.

Previous proposals from the literature used ratings and reviews as training data. However, only some clients proportionate that information after their purchases. This makes the implementation of recommendation systems in real delivery food applications difficult. This research's main contribution is using the number of orders as input for the recommendation system. These data are always stored and available in the delivery food

applications. Therefore, our methodology enables the implementation of a restaurant-recommendation system in real-life scenarios.

The research of new restaurant-recommendation systems is very important. Many consumers tend only to order in restaurants they know because they fear disappointment and do not explore other options. Therefore, this recommendation system is vital for consumers and restaurants since it can give an excellent suggestion on where to order next with high accuracy based only on clients' previous orders. The recommendations will satisfy the clients, and restaurants can increase their sales.

The future scope of this research can involve the implementation of this methodology in other areas. For example, recommendations for online sales or bookstores. Likewise, the system could be evaluated in a larger-scale real-world study to further validate its effectiveness in practice.

**Author Contributions:** Conceptualization, C.N.S. and J.D.-S.; methodology, C.N.S. and J.D.-S.; software, C.N.S. and M.G.; validation, C.N.S. and A.A.; formal analysis, C.N.S. and M.G.; investigation, C.N.S. and J.D.-S.; resources, A.A.; data curation, A.A.; writing—original draft preparation, C.N.S.; writing—review and editing, C.N.S. and J.D.-S.; visualization, C.N.S.; supervision, J.D.-S. and M.G.; project administration, J.D.-S.; funding acquisition, J.D.-S. and A.A. All authors have read and agreed to the published version of the manuscript.

**Funding:** This research was funded by Universidad Panamericana through the grant "Fomento a la Investigación UP 2019", under Project Code UP-CI-2019-DNA-AGS.

**Informed Consent Statement:** Not applicable.

**Data Availability Statement:** The data presented in this study are openly available in our GitHub repository: https://github.com/ClaudiaSanchez/DeliveryFoodApplication_RecommendationSystem, accessed on 15 January 2023.

**Acknowledgments:** Authors thank Torus Tecnologías SAPI de CV for providing the real dataset. In addition, the authors thank Samantha Licea Domínguez for helping to organize the related work.

**Conflicts of Interest:** The authors declare no conflict of interest.

## Appendix A

Table A1 shows an example from the original data: Client 463 and his/her five nearest neighbors. It was calculated using the data from 2020. Only 60 restaurants appear in the table. This is because the others have zero orders for those clients. Based on the difference metric for calculating the neighbors, we expected that near clients like the same restaurants. The preferred restaurants (with values greater than 0.1) of Client 463 are 60,186, 98,364, and 153,678. We can observe that the neighbors of Client 463 also like Restaurants 60186 and 153,678. Clients 743, 6224, and 185,054 like Restaurant 60,186, and Clients 15,413 and 1265 like Restaurant 153,678.

**Table A1.** An example from the original data of a client's normalized vector and his/her 5 nearest neighbors. Rows and columns represent clients and restaurants, respectively. Cells values mean the percentage of the orders realized by the client to the restaurant. The bluer the cell, the bigger the value.

| | 18,480 | 18,984 | 19,740 | 24,276 | 24,906 | 25,032 | 26,418 | 27,174 | 27,678 | 28,308 | 32,340 | 32,718 |
|---|---|---|---|---|---|---|---|---|---|---|---|---|
| **463** | 0.021 | 0.000 | 0.010 | 0.010 | 0.000 | 0.000 | 0.000 | 0.000 | 0.000 | 0.021 | 0.010 | 0.000 |
| **743** | 0.000 | 0.032 | 0.000 | 0.097 | 0.048 | 0.000 | 0.000 | 0.016 | 0.065 | 0.000 | 0.000 | 0.081 |
| **6224** | 0.000 | 0.075 | 0.275 | 0.000 | 0.025 | 0.000 | 0.000 | 0.025 | 0.000 | 0.000 | 0.000 | 0.000 |
| **15,413** | 0.000 | 0.000 | 0.070 | 0.000 | 0.000 | 0.000 | 0.000 | 0.000 | 0.023 | 0.000 | 0.000 | 0.000 |
| **1265** | 0.000 | 0.000 | 0.058 | 0.000 | 0.000 | 0.101 | 0.000 | 0.014 | 0.000 | 0.000 | 0.000 | 0.000 |
| **185,054** | 0.000 | 0.022 | 0.022 | 0.000 | 0.000 | 0.000 | 0.022 | 0.000 | 0.000 | 0.000 | 0.000 | 0.000 |

**Table A1.** *Cont.*

|  | 33,348 | 39,522 | 40,908 | 43,428 | 47,712 | 48,216 | 48,972 | 51,618 | 52,122 | 57,792 | 60,186 | 64,722 |
|---|---|---|---|---|---|---|---|---|---|---|---|---|
| **463** | 0.000 | 0.000 | 0.000 | 0.063 | 0.031 | 0.000 | 0.010 | 0.021 | 0.031 | 0.000 | 0.240 | 0.000 |
| **743** | 0.000 | 0.065 | 0.000 | 0.016 | 0.016 | 0.016 | 0.000 | 0.000 | 0.097 | 0.000 | 0.161 | 0.032 |
| **6224** | 0.000 | 0.000 | 0.000 | 0.000 | 0.075 | 0.000 | 0.000 | 0.000 | 0.025 | 0.000 | 0.125 | 0.000 |
| **15,413** | 0.023 | 0.047 | 0.000 | 0.000 | 0.000 | 0.000 | 0.000 | 0.000 | 0.000 | 0.000 | 0.023 | 0.000 |
| **1265** | 0.000 | 0.000 | 0.043 | 0.000 | 0.000 | 0.000 | 0.000 | 0.029 | 0.043 | 0.000 | 0.000 | 0.000 |
| **185,054** | 0.000 | 0.000 | 0.000 | 0.000 | 0.000 | 0.000 | 0.000 | 0.000 | 0.000 | 0.022 | 0.067 | 0.111 |

|  | 66,234 | 67,494 | 67,998 | 69,510 | 74,298 | 74,676 | 76,818 | 77,322 | 80,094 | 80,598 | 80,724 | 84,126 |
|---|---|---|---|---|---|---|---|---|---|---|---|---|
| **463** | 0.021 | 0.000 | 0.000 | 0.000 | 0.000 | 0.000 | 0.000 | 0.000 | 0.031 | 0.000 | 0.010 | 0.000 |
| **743** | 0.000 | 0.000 | 0.000 | 0.000 | 0.032 | 0.000 | 0.048 | 0.000 | 0.000 | 0.000 | 0.000 | 0.000 |
| **6224** | 0.000 | 0.000 | 0.000 | 0.025 | 0.000 | 0.000 | 0.175 | 0.000 | 0.000 | 0.000 | 0.000 | 0.000 |
| **15,413** | 0.000 | 0.000 | 0.000 | 0.000 | 0.116 | 0.000 | 0.000 | 0.116 | 0.000 | 0.000 | 0.000 | 0.047 |
| **1265** | 0.000 | 0.029 | 0.029 | 0.000 | 0.000 | 0.058 | 0.000 | 0.000 | 0.000 | 0.014 | 0.000 | 0.014 |
| **185,054** | 0.000 | 0.089 | 0.000 | 0.000 | 0.022 | 0.000 | 0.000 | 0.000 | 0.022 | 0.000 | 0.044 | 0.000 |

|  | 85,134 | 93,828 | 94,458 | 96,726 | 98,364 | 102,396 | 102,522 | 130,116 | 133,140 | 135,786 | 136,416 | 137,802 |
|---|---|---|---|---|---|---|---|---|---|---|---|---|
| **463** | 0.010 | 0.021 | 0.000 | 0.000 | 0.167 | 0.000 | 0.000 | 0.000 | 0.000 | 0.000 | 0.000 | 0.000 |
| **743** | 0.000 | 0.000 | 0.000 | 0.000 | 0.016 | 0.048 | 0.000 | 0.000 | 0.032 | 0.048 | 0.000 | 0.000 |
| **6224** | 0.000 | 0.000 | 0.000 | 0.000 | 0.025 | 0.000 | 0.000 | 0.000 | 0.000 | 0.000 | 0.000 | 0.000 |
| **15,413** | 0.000 | 0.000 | 0.000 | 0.000 | 0.000 | 0.000 | 0.000 | 0.000 | 0.000 | 0.000 | 0.000 | 0.140 |
| **1265** | 0.000 | 0.000 | 0.014 | 0.029 | 0.043 | 0.000 | 0.072 | 0.000 | 0.000 | 0.000 | 0.029 | 0.014 |
| **185,054** | 0.022 | 0.000 | 0.044 | 0.000 | 0.000 | 0.000 | 0.000 | 0.067 | 0.067 | 0.000 | 0.000 | 0.000 |

|  | 138,054 | 144,732 | 153,678 | 158,340 | 159,726 | 159,852 | 161,868 | 164,010 | 166,278 | 184,926 | 186,690 | 201,432 |
|---|---|---|---|---|---|---|---|---|---|---|---|---|
| **463** | 0.000 | 0.000 | 0.146 | 0.010 | 0.000 | 0.000 | 0.000 | 0.031 | 0.000 | 0.031 | 0.021 | 0.021 |
| **743** | 0.000 | 0.000 | 0.016 | 0.000 | 0.000 | 0.000 | 0.000 | 0.000 | 0.016 | 0.000 | 0.000 | 0.000 |
| **6224** | 0.000 | 0.000 | 0.025 | 0.000 | 0.000 | 0.000 | 0.000 | 0.000 | 0.000 | 0.025 | 0.100 | 0.000 |
| **15,413** | 0.000 | 0.000 | 0.116 | 0.047 | 0.023 | 0.000 | 0.163 | 0.000 | 0.000 | 0.047 | 0.000 | 0.000 |
| **1265** | 0.000 | 0.058 | 0.159 | 0.000 | 0.000 | 0.014 | 0.000 | 0.130 | 0.000 | 0.000 | 0.000 | 0.000 |
| **185,054** | 0.356 | 0.000 | 0.000 | 0.000 | 0.000 | 0.000 | 0.000 | 0.000 | 0.000 | 0.000 | 0.000 | 0.000 |

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
