# Peer review of "Recommendation System for a Delivery Food Application Based on Number of Orders"

_applsci, doi:10.3390/app13042299_

Round 1
Reviewer 1 Report
The paper is well-motivated. I have some minor suggestions for the authors to improve the paper further.
(1) The abstract should be improved. The research gap between the existing literature and the paper’s main contribution must be clearly stated without digressing from the gist of the paper. I recommend analyzing and adding to the reference the following works: doi: 10.1007/s12652-019-01203-7, doi: 10.3390/jintelligence10020032 , doi: 10.1007/s00607-018-0680-z .
(2) There are many papers in this research area, and it is unclear whether the authors collected these papers based on which criteria. There is also some missing information in these papers. Please calibrate the reference section carefully.
(3) Provide a table to analyze the literature review. What are the research gaps? How can you fill them? Compare the existing models in the literature and why you think your model is merit in the literature. Please work on improving the clarity of your paper.
Reviewer 2 Report
The idea is interesting and article is well written. However, following modifications are suggested to further improve the quality of the submission.
1. The introduction section lacks the discussion about the recent recommendation systems based on fractional calculus concepts. In this regard, please see the profile of Prof. Zeshan Aslam Khan.
2. The main contribution of the current study needs to be highlighted in the introduction section.
3. The mathematical description of the proposed scheme is missing.
4. KNN is well established approach. Explain the KNN in detail with some mathematics regarding the current study.
5. Please provide the elaborative mathematical description of the evaluation metrics.
6. Rewrite the conclusions to reflect the main findings of the study.
Reviewer 3 Report
The following should be noted and corrected accordingly:
1. How practicable is your proposed model in real-time?
2. Is it cost-efficient?
3. Some diagrams and terms are not properly explained
4. Grammar requires minor re-editing
5. Are the numbers and formulas here generic or generated by you?
6. What is the future scope of the study?
7. Consider the following related papers to embellish your paper:
• https://doi.org/10.1142/S0219622021500619
• doi: 10.1109/ACCESS.2020.2968537.
Round 2
Reviewer 2 Report
The revised manuscript can be accepted for publication.